# Healthcare workers' knowledge, attitude and practices during the COVID-19 pandemic response in a tertiary care hospital of Nepal

Dil K. Limbu[1], Rano M. Piryani[2], Avinash K. Sunny [3]*

1 Deparment of General Practice and Emergency Medicine, Universal College of Medical Sciences, Bhairahawa, Nepal, 2 Department of Internal Medicine, Universal College of Medical Sciences, Bhairahawa, Nepal, 3 Research Division, Golden Community, Lalitpur, Nepal

* avinashkayastha@gmail.com

**Data Availability Statement:** All relevant data are within the manuscript and its Supporting Information files.

## Abstract

### Background

COVID-19 is an ongoing pandemic, for which appropriate infection prevention and control measures need to be adopted. Healthcare workers' adherence to prevention and control measures is affected by their knowledge, attitudes, and practices (KAP) towards COVID-19. In this study, we assessed the KAP among healthcare workers towards the COVID-19 during the ongoing pandemic.

### Method

A self-developed piloted KAP questionnaire was administered to the recruited healthcare workers involved in the COVID-19 response at the Universal College of Medical Sciences Teaching Hospital (UCMSTH), in Bhairahawa, Nepal. The knowledge questionnaire consisted of questions regarding the clinical characteristics, prevention, and management of COVID-19. Assessment on attitudes and practices towards COVID-19 included questions on behaviour and change in practices made towards COVID-19 response. Knowledge scores were calculated and compared by demographic characteristics and their attitude and practices towards COVID-19. Data were analysed using bivariate statistics.

### Results

A total of 103 healthcare workers participated in the study. The mean age of the participants was 28.24±6.11 years (range: 20–56); 60.2% were females; 61.2% were unmarried; 60.2% had a medical degree, and 39.8% were the nursing staff. The mean knowledge score was 10.59±1.12 (range: 7–13), and it did not vary significantly when adjusted for demographic characteristics. The attitude was positive for 53.4% of the participants with a mean knowledge score of 10.35±1.19 and negative for 46.6% participants with a mean knowledge score of 10.88±0.98 (p = 0.02). The practice was good (≥3 score) for 81.5% participants with a mean knowledge score of 10.73±1.12 and poor for 18.5% participants with a mean

**Funding:** The author(s) received no specific funding for this work.

**Competing interests:** The authors have declared that no competing interests exist.

knowledge score of 10.46±1.13 (p = 0.24). The attitude of the participants improved with increasing age (29.55±7.17, p = 0.02).

## Conclusion

There is comparably better knowledge regarding COVID-19 among healthcare workers. Appropriate practice correlates with better knowledge and positive attitude towards COVID-19 infection is seen with increasing age. Hence, training on protection and protective measures for having a positive attitude among healthcare workers is necessary against the fight with COVID-19 infection.

## Introduction

COVID-19 is a disease caused by the SARS-CoV-2 virus, first identified in the city of Wuhan, in China's Hubei province in December 2019 [1]. COVID-19 was previously known as the 2019 novel Corona virus (2019-nCoV) respiratory disease before the World Health Organization (WHO) declared the official name as COVID-19 in February 2020 [2]. On March 11, 2020, the World Health Organization (WHO) declared the COVID-19 outbreak a pandemic [3]. This ongoing pandemic has been spreading very rapidly, with more than 8.5 million confirmed infections and more than 0.47 million deaths worldwide as of June 22 2020 (GMT 01.18) [4].

Countries worldwide have used various control measures such as social distancing, hand washing, shutting public transportation and public places, and finally testing and tracing affected communities [5]. Like many governments around the world, Nepal has also called for lockdown since 24[th] March, 2020, allowing only essential services like hospitals, groceries, and medical supplies and frontline emergency services [6]. In Nepal, the total number of confirmed infections stood at 9561 as of June 22, 2020 [7].

Profound knowledge supports an optimistic attitude and appropriate practices at work, which helps deter the risk of infection [8]. Healthcare workers' adherence to control measures is affected by their knowledge, attitudes, and practices (KAP) towards COVID-19. Therefore, it is crucial to understand the knowledge of the medical providers and determine the factors that affect their attitudes and practices to have adequate practices and protection. Thus, this study aimed to assess the KAP among healthcare workers towards COVID-19 infections during the ongoing pandemic.

## Method

### Study design and setting

A cross-sectional study was conducted from April 7, 2020 to May 7, 2020 at Universal College of Medical Sciences Teaching Hospital (UCMSTH), Bhairahawa, Nepal. The hospital has recently stepped up its efforts to support the government in its fight towards COVID 19. UCMSTH is a teaching hospital serving as a tertiary care centre located in the southwest border of Nepal. All the healthcare providers working in the hospital during the COVID-19 pandemic response were included in the study.

## Data collection and management

Data were collected from healthcare workers using a self-administered questionnaire to assess KAP towards COVID-19. This questionnaire consisted of two parts: the demographic information and a self-developed piloted and pretested KAP questionnaire. The KAP questionnaire assessed knowledge, attitude and practices of COVID-19 response. Knowledge was measured using 13 true and false questions. These questions assessed the provider's knowledge on clinical manifestations, mode of transmission, prevention and control of COVID-19. The reliability analysis of the knowledge questionnaire with Cronbach's alpha coefficient was accep with an internal consistency of 0.76. Attitude and practice were assessed using 5 questions with a yes/no/don't know option. (Table 1)

Collected data was then entered into an MS Excel sheet and coded for anonymity. The entered data was exported into Statistical Package for Social Sciences (SPSS) version 23 for analysis.

**Table 1. Correct responses on KAP questionnaire (n = 103).**

| Knowledge Questions | Correct response (%) |
|---|---|
| 1. Fever, dry cough, difficulty in breathing, tiredness are the common clinical symptoms of COVID-19. | 103 (100%) |
| 2. Sneezing, runny nose, stuffy nose and headache are less common in persons infected with COVID-19. | 80 (77.7%) |
| 3. Loss of taste and smell are also the feature of COVID 19 infection. | 52 (50.5%) |
| 4. Currently there is no treatment of COVID-19 infection, but early symptomatic and supportive treatment can help most patient recover from infection. | 101 (98.1%) |
| 5. Majority of COVID-19 infective patient will not develop severe illness but elderly, patient having chronic illness, DM, COPD are likely to develop severe illness. | 86 (83.5%) |
| 6. COVID-19 infected person with fever can infect to other people. | 28 (27.2%) |
| 7. COVID-19 virus spread via respiratory droplets. | 99 (96.1%) |
| 8. Ordinary people should wear general mask. | 85 (82.5%) |
| 9. People maintain 2-meter distance in the public places. | 94 (91.3%) |
| 10. Lockdown is effective measure to slow the spread of infection. | 102 (99.0%) |
| 11. People infected with COVID-19 should immediately place in proper isolation. | 102 (99.0%) |
| 12. Suspected COVID19 patient should be sent to a quarantine centre or home quarantine. | 99 (96.1%) |
| 13. Health care professional with direct contact should take tablet hydroxychloroquine as a prophylaxis. | 60 (58.3%) |
| **Attitude Questions** | **Correct response (%)** |
| 1. Can Nepal win the battle against COVID-19? | 52 (50.5%) |
| 2. Are you confident to work in hospital during COVID-19 pandemic? | 47 (45.6%) |
| 3. Does your family support you to work in hospital during pandemic? | 58 (56.3%) |
| 4. Do you experience anxiety and fear while working with suspected COVID-19 patient? | 81 (78.6%) |
| 5. Have all the doctors from various department actively involved in COVID-19 Pandemic response? | 38 (36.9%) |
| **Practice Questions** | **Correct response (%)** |
| 1. Are you being trained to work for COVID-19 patient? | 18 (17.5%) |
| 2. Have you following social distancing? | 78 (75.5%) |
| 3. Have you been wearing mask and gloves during hospital practice? | 99 (96.1%) |
| 4. Do you regularly follow infection protection measures? | 87 (84.5%) |
| 5. Are you attending patient suspected with COVID-19? | 71 (68.9%) |

## Study variables

***Demographic variables*** included age as a continuous variable; gender as male and female; marital status as married and unmarried. For education, Auxiliary Nurse Midwife (ANM), Proficiency Certificate Level (PCL) Nursing and Community Medical Assistant (CMA) were included as certificate level; Bachelor in Nursing Science (BNS) as graduate level in nursing; Bachelor of Medicine and Bachelor of Surgery (MBBS) as graduate medical education; Doctor of Medicine (MD), Master in Dental Surgery (MDS) and Doctorate of Medicine (DM) as post-graduate medical education. Designation were included as Nursing staff, Medical Intern, Medical Officer, Post Graduate (PG) Resident, and Consultant.

***Knowledge scores*** were calculated by assigning 1 point to each correct answer, and a 0 to an incorrect/unknown answer. The total knowledge score ranged from 0 to 13, with higher scores signifying better knowledge.

***Attitude*** was assessed as positive and negative. The average score on the attitude questionnaire was calculated and used as a cut off for positive and negative. The average score was 3 and scores below 3 was labelled negative.

***Practice*** was assessed as good and poor. The average score on the practice questionnaire was calculated and used as a cut off for good and poor. The average score was 3 and scores below 3 was labelled poor.

## Statistical analysis

Descriptive statistics were calculated as frequency, percentage, mean and standard deviation (SD). Data were analysed using Pearson chi-square test, Pearson correlation, independent t-test, and one way analysis of variance (ANOVA) test. At 95% Confidence Interval, p-value $< 0.05$ was considered to be statistically significant.

## Ethical approval

Ethical clearance was obtained from the Institutional Review Committee (IRC) of Universal College of Medical Sciences Teaching Hospital (UCMSTH). Informed written consents were taken from the participant before inclusion in the study, and confidentiality was maintained throughout.

# Result

The correct responses to KAP are mentioned in Table 1. A total of 103 healthcare workers participated in the study with a male to female ratio of 1:1.5. The mean age of the participants was 28.24±6.11 years (range: 20–56), 62 (60.2%) were females, 63 (61.2%) were unmarried, 62 (60.2%) had a medical degree and 41 (39.8%) were the nursing staff. The mean knowledge score was 10.59±1.12 (range: 7–13) with 81.5% correct answer rate. There was no significant correlation in the participants' knowledge score and their mean age (p = 0.13). The mean knowledge scores for male participants was 10.76±1.16, while for female participants was 10.48 ±1.10, with no statistical significance (p = 0.24). Married participants had a mean knowledge score of 10.78±1.00, while for unmarried participants, it was 10.48±1.19, suggesting no statistically significant difference (p = 0.19). By education, the mean knowledge score was highest for MBBS education with 10.85±1.28 (p = 0.06), and by designation; it was highest for Resident with 11.00±0.86 (p = 0.05) with no statistical significance. (Table 2)

The mean knowledge score varied significantly with the attitude of the participants (p = 0.02) but it did not vary with their practice (p = 0.24). Attitude was positive ($\geq$3 score) for 55(53.4%) participants with a mean knowledge score of 10.35±1.19 and negative ($<$3 score)

**Table 2. Knowledge scores of COVID-19 by demographic variables (n = 103).**

| Characteristics | Category | N (%) | Knowledge score (Mean ± SD) | p-value |
|---|---|---|---|---|
| **Age (years)** | 28.24±6.11 | 103(100) | 10.59±1.12 | 0.13[a] |
| **Gender** | Male | 41(39.8) | 10.76±1.16 | 0.24[b] |
| | Female | 62(60.2) | 10.48±1.10 | |
| **Marital status** | Unmarried | 63(61.2) | 10.48±1.19 | 0.19[b] |
| | Married | 40(38.8) | 10.78±1.00 | |
| **Education** | ANM/PCL/CMA | 34(33.0) | 10.18±1.03 | 0.06[c] |
| | BNS | 7(6.8) | 10.57±0.96 | |
| | MBBS | 47(45.6) | 10.85±1.28 | |
| | MD/MDS/DM | 15(14.6) | 10.73±1.12 | |
| **Designation** | Nursing staff | 41(39.8) | 10.24±1.18 | 0.05[c] |
| | Medical Intern | 23(22.3) | 10.87±0.92 | |
| | Medical Officer | 4(3.9) | 10.00±1.41 | |
| | Resident | 20(19.4) | 11.00±0.86 | |
| | Consultant | 15(14.6) | 10.73±1.28 | |
| **Attitude** | Positive (≥3) | 55(53.4) | 10.35±1.19 | **0.02[d]** |
| | Negative (<3) | 48(46.6) | 10.88±0.98 | |
| **Practice** | Good (≥3) | 84(81.5) | 10.73±1.12 | 0.24[d] |
| | Poor (<3) | 19(18.5) | 10.46±1.13 | |

[a]Pearson correlation

[b]Independent t-test

[c]one way ANOVA test

[d]Pearson chi-square test.

for 48(46.6%) participants with a mean knowledge score of 10.88±0.98. Similarly, practice was good (≥3 score) for 84(81.5%) participants with a mean knowledge score of 10.73±1.12 and practice was poor (<3 score) for 19(18.5%) participants with a mean knowledge score of 10.46 ±1.13. There was a negative correlation between knowledge and attitude (r = -0.313, p = 0.001), however, knowledge didn't correlate with practice (r = 0.093, p = 0.35). There was a significant correlation between attitude and practice (r = 0.298, p = 0.002). (Tables 2 & 3)

Association of attitude and practice of the participants towards COVID-19 with their characteristics were assessed. There was no significant difference in the attitude of the participants by their gender (p = 0.66), marital status (p = 0.51), education (p = 0.42) or designation (p = 0.28); however, the attitude varied significantly with the increasing age of the participants (p = 0.02). The practice of the participants was mostly good and did not differ by age

**Table 3. Correlation between the knowledge, attitude and practice scores (n = 103).**

| Variable (Mean ± SD) | | Knowledge | Attitude | Practice |
|---|---|---|---|---|
| **Knowledge** (10.59±1.12) | Correlation Coefficient (r) | 1 | -0.313 | 0.093 |
| | p-value | | **0.001***| 0.350* |
| **Attitude** (2.68±1.16) | Correlation Coefficient (r) | -0.313 | 1 | 0.298 |
| | p-value | **0.001*** | | **0.002*** |
| **Practice** (3.43±1.03) | Correlation Coefficient (r) | 0.093 | 0.298 | 1 |
| | p-value | 0.350* | **0.002*** | |

*Pearson correlation.

**Table 4. Attitude and practice towards COVID-19 by demographic variables (n = 103).**

| Characteristics | Category | Positive attitude N (%) | p-value | Good Practice N (%) | p-value |
|---|---|---|---|---|---|
| **Age (years)** | | 29.55±7.17 | **0.02**[*] | 28.38±6.10 | 0.64[*] |
| **Gender** | Male | 23(41.8) | 0.66 | 33(39.3) | 0.82 |
| | Female | 32(58.2) | | 51(60.7) | |
| **Marital status** | Unmarried | 32(58.2) | 0.51 | 50(59.5) | 0.47 |
| | Married | 23(41.8) | | 34(40.5) | |
| **Education** | ANM/PCL/CMA | 18(32.7) | 0.42 | 26(31.0) | 0.78 |
| | BNS | 2(3.6) | | 6(7.1) | |
| | MBBS | 25(45.5) | | 40(47.6) | |
| | MD/MDS/DM | 10(18.2) | | 12(14.3) | |
| **Designation** | Nursing | 20(36.4) | 0.28 | 32(38.1) | 0.09 |
| | Medical Intern | 9(16.4) | | 16(19.0) | |
| | Medical Officer | 3(5.5) | | 4(4.8) | |
| | Resident | 13(23.6) | | 20(23.8) | |
| | Consultant | 10(18.2) | | 12(14.3) | |

[*]Independent t-test.

(p = 0.64), gender (p = 0.82), marital status (p = 0.47), education (p = 0.78) or designation (p = 0.09). (Table 4)

## Discussion

This is a cross-sectional study conducted with the objective to assess the KAP among healthcare workers toward the ongoing COVID-19 pandemic. The findings of this study with relation to 81.5% providers answering correctly on the knowledge questionnaire is comparable with a similar study conducted in China, which reported that 89% healthcare workers surveyed demonstrated sufficient knowledge on COVID-19 [9]. As this study was conducted during the national lockdown period in Nepal, healthcare workers were quite aware of most of the information related to COVID-19 as part of being prepared to respond to the ongoing pandemic. Despite the differences in healthcare workers' demographic characteristics, the knowledge seemed to be on par with all of them. This is in contrast with other studies suggesting differences in knowledge with differing types of healthcare workers [10].

Overall, 53.4% of the healthcare workers had a positive attitude towards the COVID-19. This finding is lower compared to that in other studies conducted in China. This attitude could probably be due to only half of medical providers (50.5%) belief that Nepal could win the fight against COVID-19. Even with having family support (56.3%), they were less confident (45.6%) while at work because of increased anxiety and fear (78.6%). The attitude of the healthcare workers was found to be similar across different demographic characteristics. This suggests that the notion was the same for all in this pandemic situation, defying the reports from other studies of varying attitudes by the healthcare workers' demographics [11].

Having better knowledge of COVID-19 among the healthcare workers did not correlate with their attitude towards the disease while at work in this study. This finding is in contrast with other studies reporting that knowledge directly affected their attitude and increased their confidence [9, 10]. Knowledge is a prerequisite for promoting preventive measures and forming positive attitudes towards the fight against the disease [8]. However, studies have reported that protective measures not being in place could increase the chances of infection, while well-protected emergency and other departments in the hospital had lower chances of infection

[12]. Healthcare workers with higher age elicited positive attitudes in this study. Similar was the case among healthcare workers in other studies. The higher age, the longer is the experience in dealing with emergencies, ultimately demonstrating confidence and optimism. Hence, increasing age could be the reason for a positive attitude [13].

In this study, healthcare workers' practices were found to be mostly appropriate, and it did not differ by their demographic characteristics or knowledge scores. However, the practice significantly correlated with their attitude. Thus, poor practices can be linked to poor attitude, and it resulted so because very few (17.5%) were trained to work for COVID-19 patient, despite many following practices such as social distancing (75.5%), wearing mask and gloves during hospital practice (96.1%), infection protection measures (84.5%) and attending patient suspected with COVID-19 (68.9%). This finding is comparable to a similar study in China, which found 89.7% of the healthcare workers followed correct practices regarding COVID-19 [9]. Moreover, they cannot neglect their protection by engaging in best practices at work as they are the most vulnerable to infection [14].

## Limitations

The findings of this study should be cautiously used for generalization since it depicts one hospital in Nepal. Additionally, practices being self-reported may not be actual; hence further study is warranted.

## Conclusion

This study found out that there is a positive correlation between knowledge regarding COVID-19 among healthcare workers and appropriate clinical practices. However, their attitude was less optimistic even with better knowledge. Higher optimism was seen with healthcare workers' higher age. Healthcare workers practice is directly correlated with their attitude. Hence, despite better knowledge, there is a need for a more positive attitude at the place of practice. Also, education and training on protection and protective measures are required to improve positive attitude and better practices at work during the COVID-19 pandemic response.

## Supporting information

**S1 File.**
(XLSX)

## Acknowledgments

We thank all the healthcare workers for their voluntary participation in the study.

## Author Contributions

**Conceptualization:** Dil K. Limbu, Avinash K. Sunny.

**Formal analysis:** Dil K. Limbu, Rano M. Piryani, Avinash K. Sunny.

**Investigation:** Dil K. Limbu.

**Methodology:** Dil K. Limbu, Avinash K. Sunny.

**Supervision:** Rano M. Piryani, Avinash K. Sunny.

**Writing – original draft:** Dil K. Limbu, Avinash K. Sunny.

**Writing – review & editing:** Dil K. Limbu, Rano M. Piryani, Avinash K. Sunny.

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
