## [Decision Letter · Decision Letter 0]

2 Oct 2020

PONE-D-20-25614

Healthcare workers’ knowledge, attitude and practices during the COVID-19 pandemic response in a tertiary care hospital of Nepal

PLOS ONE

Dear Dr. Sunny,

Thank you for submitting your manuscript to PLOS ONE. After careful consideration, we feel that it has merit but does not fully meet PLOS ONE’s publication criteria as it currently stands. Therefore, we invite you to submit a revised version of the manuscript that addresses the points raised during the review process.

We look forward to receiving your revised manuscript.

Kind regards,

Amit Sapra

Academic Editor

PLOS ONE

Journal Requirements:

Additional Editor Comments (if provided):

The manuscript is very interesting and pertinent to current times. Please view the reviewer's comments to make the recommended changes before the publication is approved.

Reviewers' comments:

Reviewer's Responses to Questions

**Comments to the Author**

1. Is the manuscript technically sound, and do the data support the conclusions?

Reviewer #1: Yes

Reviewer #2: Yes

2. Has the statistical analysis been performed appropriately and rigorously? 

Reviewer #1: Yes

Reviewer #2: Yes

3. Have the authors made all data underlying the findings in their manuscript fully available?

Reviewer #1: Yes

Reviewer #2: Yes

4. Is the manuscript presented in an intelligible fashion and written in standard English?

Reviewer #1: Yes

Reviewer #2: No

5. Review Comments to the Author

Reviewer #1: Please see attached Document for recommended changes. There were many Grammar and Syntax errors that I corrected. It is a well done study and very appropriate in today's times. I enjoyed reading it. Great job!

Reviewer #2: Please review my edits. Too many grammatical errors. Data is presented in a very complex way and too many running sentences. Please view the document with enabling the edits so you can see the revisions requested. Read my questions in () sentences to elaborate.

6. PLOS authors have the option to publish the peer review history of their article (what does this mean?). If published, this will include your full peer review and any attached files.

Reviewer #1: **Yes: **Priyanka Bhandari

Reviewer #2: **Yes: **Waiz A Wasey

---

## [Author Response · Author response to Decision Letter 0]

9 Oct 2020

Reviewer #1: Please see attached Document for recommended changes. There were many Grammar and Syntax errors that I corrected. It is a well done study and very appropriate in today's times. I enjoyed reading it. Great job!

Response: Thank you so much for appreciation and corrections as well. I have revised the corrections accordingly.

Reviewer #2: Please review my edits. Too many grammatical errors. Data is presented in a very complex way and too many running sentences. Please view the document with enabling the edits so you can see the revisions requested. Read my questions in () sentences to elaborate.

Response: Thank you so much for the review and suggestions. I have constructed simpler sentences as suggested and elaborated at the mentioned places.

---

## [Editor Report · Decision Letter 1]

28 Oct 2020

Healthcare workers’ knowledge, attitude and practices during the COVID-19 pandemic response in a tertiary care hospital of Nepal

PONE-D-20-25614R1

Dear Dr. Sunny

We’re pleased to inform you that your manuscript has been judged scientifically suitable for publication and will be formally accepted for publication once it meets all outstanding technical requirements.

Kind regards,

Amit Sapra

Academic Editor

PLOS ONE
---

## [Editor Report · Acceptance letter]

29 Oct 2020

PONE-D-20-25614R1 

Healthcare workers’ knowledge, attitude and practices during the COVID-19 pandemic response in a tertiary care hospital of Nepal 

Dear Dr. Sunny:

I'm pleased to inform you that your manuscript has been deemed suitable for publication in PLOS ONE. Congratulations! Your manuscript is now with our production department. 

Kind regards, 

on behalf of

Dr. Amit Sapra 

Academic Editor

PLOS ONE